# Anti-Diabetic Activity of 2,3,6-Tribromo-4,5-Dihydroxybenzyl Derivatives from *Symphyocladia latiuscula* through PTP1B Downregulation and α-Glucosidase Inhibition

**DOI:** 10.3390/md17030166

**Published:** 2019-03-14

**Authors:** Pradeep Paudel, Su Hui Seong, Hye Jin Park, Hyun Ah Jung, Jae Sue Choi

**Affiliations:** 1Department of Food and Life Science, Pukyong National University, Busan 48513, Korea; phr.paudel@gmail.com (P.P.); seongsuhui@naver.com (S.H.S.); 2Department of Food Science and Nutrition, Changshin University, Gyeongsangnam-do, Changwon 51352, Korea; parkhj@cs.ac.kr; 3Department of Food Science and Human Nutrition, Chonbuk National University, Jeonju 54896, Korea

**Keywords:** *Symphyocladia latiuscula*, bromophenols, PTP1B, insulin-resistant HepG2, diabetes

## Abstract

The marine alga, *Symphyocladia latiuscula* (Harvey) Yamada, is a good source of bromophenols with numerous biological activities. This study aims to characterize the anti-diabetic potential of 2,3,6-tribromo-4,5-dihydroxybenzyl derivatives isolated from *S. latiuscula* via their inhibition of tyrosine phosphatase 1B (PTP1B) and α-glucosidase. Additionally, this study uses in silico modeling and glucose uptake potential analysis in insulin-resistant (IR) HepG2 cells to reveal the mechanism of anti-diabetic activity. This bioassay-guided isolation led to the discovery of three potent bromophenols that act against PTP1B and α-glucosidase: 2,3,6-tribromo-4,5-dihydroxybenzyl alcohol (**1**), 2,3,6-tribromo-4,5-dihydroxybenzyl methyl ether (**2**), and bis-(2,3,6-tribromo-4,5-dihydroxybenzyl methyl ether) (**3**). All compounds inhibited the target enzymes by 50% at concentrations below 10 μM. The activity of **1** and **2** was comparable to ursolic acid (IC_50_; 8.66 ± 0.82 μM); however, **3** was more potent (IC_50_; 5.29 ± 0.08 μM) against PTP1B. Interestingly, the activity of **1**–**3** against α-glucosidase was 30–110 times higher than acarbose (IC_50_; 212.66 ± 0.35 μM). Again, **3** was the most potent α-glucosidase inhibitor (IC_50_; 1.92 ± 0.02 μM). Similarly, **1**–**3** showed concentration-dependent glucose uptake in insulin-resistant HepG2 cells and downregulated PTP1B expression. Enzyme kinetics revealed different modes of inhibition. In silico molecular docking simulations demonstrated the importance of the 7–OH group for H-bond formation and bromine/phenyl ring number for halogen-bond interactions. These results suggest that bromophenols from *S. latiuscula*, especially highly brominated **3**, are inhibitors of PTP1B and α-glucosidase, enhance insulin sensitivity and glucose uptake, and may represent a novel class of anti-diabetic drugs.

## 1. Introduction

Diabetes is a group of metabolic diseases characterized by high blood sugar levels. As the disease progresses, it can lead to blurred vision, cardiovascular diseases, kidney and/or other organ damage and dysfunction, and even depression [1]. Diabetes is caused by a failure of the pancreas to produce insulin (type I diabetes) or by insulin resistance (cells do not respond to insulin; type II diabetes). Type II diabetes mellitus (T2DM) accounts for more than 90% of total diabetes cases [2]. Protein tyrosine phosphatase 1B (PTP1B) downregulates insulin signaling by catalyzing the dephosphorylation of phosphotyrosine residues and controlling the phosphorylation levels of proteins involved in insulin signaling pathways [3]. Previous research has reported phenotypes with enhanced sensitivity to insulin and lower plasma glucose and insulin levels in PTP1B knockout mice, which implies that PTP1B is an attractive therapeutic target for T2DM [4,5]. Thus, in the past few years, the development of PTP1B inhibitors has been a focus of both the pharmaceutical industry and academia. Unfortunately, the development of small-molecule PTP1B inhibitors into effective drugs has proven difficult due to their low cell permeability and bioavailability [6]. Therefore, efforts to find more efficient PTP1B inhibitors are ongoing, with a focus on those from the marine environment (which has been called an “endless source of health promoting components” [7]).

The marine environment is a rich source of structurally diverse and unique compounds. Approximately 5000 halogenated compounds have been identified in marine sources thus far. In response to various ecological pressures, marine organisms such as algae, sponges, and corals produce a wide range of biologically active secondary metabolites, 45% of which are brome-metabolites due to the high bromine concentration in sea water [8,9]. Natural brominated compounds have been reported to exhibit a variety of biological activities including antibacterial, antifungal, antiviral, antioxidant, antitumor, anti-inflammatory, and enzymatic activity through protein kinase and acetyl-cholinesterase inhibition [10,11,12,13]. Secondary metabolites from marine sources have been regarded as an invaluable source of therapeutic agents, and these metabolites have been subsequently modified via semi-synthetic routes to enhance their therapeutic effects. Compared with chemically synthesized compounds (high efficacy with numerous adverse effects), semi-synthesized compounds developed by modifying natural secondary metabolites often have fewer side effects, thus increasing the allure of bioactive compounds from marine sources for drug discovery.

Halogenation is a traditional tool in drug optimization and is an important factor in drug discovery and development for four main reasons: (1) the degree of halogenation influences lipophilicity and cell membrane solubility; (2) halogenation improves membrane binding, permeation, and diffusion; (3) halogenation increases the half-life of pharmaceutical drugs; and (4) halogenation enhances blood-brain barrier permeability and improves central nervous system delivery [14,15,16,17]. Halogenated compounds are predominantly available from marine sources. The red alga *Symphyocladia latiuscula* (Harvey) Yamada has been reported to be a good source of bromophenols with numerous biological activities including antibacterial [18], antiviral [19], antifungal [20], anticancer [21], free radical scavenging [22], aldose reductase inhibitory [23], α-glucosidase inhibitory [24], and other properties [25,26,27]. Bromophenols from *S. latiuscula* often contain one prime 2,3,6-tribromo-4,5-dihydroxybenzyl moiety with various substituents. This study aims to discover antidiabetic brominated compounds. In support of this goal, we performed enzyme kinetics and in silico molecular modeling on the enzymes used in inhibition assay. We also evaluated insulin sensitizing potential of test compounds using 2-[*N*-(7-nitrobenz-2-oxa-1,3-diazol-4-yl) amino]-2-deoxyglucose (2-NBDG) glucose uptake assays in insulin-resistant HepG2 cells and PTP1B protein expression using the western blot technique.

## 2. Results

### 2.1. Inhibitory Activity of Methanol Extract and Solvent Soluble Fractions on Tyrosine Phosphatase 1B (PTP1B) and α-Glucosidase

The initial step in evaluating anti-diabetic secondary metabolites from *Symphyocladia latiuscula* was the evaluation of PTP1B and α-glucosidase inhibition potentials of methanol extract and fractions obtained by MeOH extract partition with different solvents using *p*NPP and *p*NPG as substrates, respectively. Enzyme inhibition levels are expressed as mean IC_50_ values with standard deviation. As shown in Table 1, methanol extract inhibited both PTP1B and α-glucosidase enzyme activity with IC_50_ values of 9.01 ± 0.33 and 91.58 ± 9.66 μg/mL, respectively. The CH_2_Cl_2_, EtOAc and *n*-BuOH solvent fractions all inhibited the activity of both the enzymes with more promising results. The EtOAc fraction showed particularly strong inhibition of both the PTP1B and α-glucosidase enzymes with IC_50_ values of 2.79 ± 0.11 and 6.71 ± 0.15 μg/mL, respectively. Because this latter fraction gave the most promising inhibition, further research was carried out to characterize potential anti-diabetic phytochemicals in the EtOAc fraction.

### 2.2. Inhibitory Activity of Bromophenols on PTP1B and α-Glucosidase

Three bromophenols were isolated from the active EtOAc fraction by using open Si gel column chromatography and purified via series of multiple Reverse Phase column chromatography. The PTP1B and α-glucosidase inhibitory activities of bromophenols **1**–**3** (Figure 1) are presented in Table 2. A PTP1B enzyme inhibition assay that was performed using ursolic acid as a reference drug (IC_50_; 8.66 ± 0.82 μM) showed that the activity of bromophenols was comparable with ursolic acid. Bromophenol **3** had an IC_50_ value of 5.29 ± 0.08 μM, making it the most active among the tested compounds, followed by **1** (IC_50_; 7.74 ± 0.14 μM) and **2** (IC_50_; 8.50 ± 0.45 μM). Similarly, the α-glucosidase inhibition assay was validated with acarbose (IC_50_; 212.66 ± 0.35 μM) as a reference drug. The tested bromophenols showed a 30–110-fold increase in α-glucosidase inhibition activity compared to acarbose. The relative α-glucosidase enzyme inhibition of the three bromophenols was similar to that of PTP1B enzyme inhibition: **3** was the most active with an IC_50_ value 1.92 ± 0.02 μM, followed by **1** (IC_50_; 2.63 ± 0.11 μM) and **2** (IC_50_; 7.24 ± 0.02 μM).

### 2.3. Enzyme Kinetics of PTP1B and α-Glucosidase Inhibition

In order to discern the mode of PTP1B and α-glucosidase inhibition by bromophenols, a kinetic study was performed at different substrate concentrations for both enzymes.

The mode of enzyme inhibition characterized by Lineweaver–Burk plots (Figure 2; Figure 3) is presented in Table 2. Compounds **1** and **2** appeared to be mixed-type inhibitors for the PTP1B enzyme (as inhibitor concentration increased, *K*_m_ increased and *V*_max_ decreased) with *K*_i_ values of 1.19 for compound **1** and 2.40 μM for compound **2**. In contrast, compound **3** appeared to be a competitive inhibitor with an inhibition constant of 2.25 μM. In competitive inhibition mode (Figure 2C), double reciprocal plots yielded a group of lines with the same *y*-intercept (constant *V*_max_) but different *x*-intercepts (varying *K*_m_ values). Similarly, **1** and **2** were both non-competitive inhibitors of α-glucosidase (as inhibitor concentration increased, *V*_max_ decreased but *K*_m_ remained constant) with dissociation constants of 1.92 and 5.54 μM, respectively (Figure 3A,B). The double reciprocal plots of α-glucosidase enzyme inhibition by **3** yielded a group of lines intersecting at a single point in the second quadrant. Upon increasing the concentration of **3**, *V*_max_ decreased and *K*_m_ increased, likely indicating a mixed-type mode of inhibition.

### 2.4. Molecular Docking Simulation of PTP1B Inhibition

The binding pose of inhibitors within the PTP1B active site (PDB ID: 1T49) was investigated via molecular docking simulation (Figure 4). The simulation study was validated using two reference inhibitors: 3-({5-[(*N*-acetyl-3-{4-[(carboxycarbonyl)(2-carboxyphenyl)amino]-1-naphthyl}-L-alanyl) amino]pentyl}oxy)-2-naphthoic acid (compound A) as a catalytic inhibitor and 3-(3,5-dibromo-4-hydroxy-benzoyl)-2-ethyl-benzofuran-6-sulfonic acid (4-sulfamoyl-phenyl)-amide (compound B) as an allosteric inhibitor. Simulation results are presented in Figure 5 and Table 3. Bromophenols **1** and **2** bound to both the catalytic and allosteric pockets of PTP1B with similar binding energies (−5.80 to −5.96 kcal/mol), forming various H-bond interactions and halogen-bond interactions via bromine atoms. The PTP1B-**1** inhibitor complex at the allosteric pocket had a −5.83 kcal/mol binding energy with three hydrogen bonds interacting with amino acid residues Ala189, Glu276, and Gly277. Beside this, other interactions with Arg221, Trp179, Gln266, Thr263, and Gly183 were observed for both inhibitors **1** and **2**. The 7-OH group of **1** displayed additional H-bond interactions with Phe182 and Lys116 at a distance of 2.0 and 2.2 Å in a catalytic pocket, respectively. However, H-bond interactions with these two residues were not observed for inhibitor **2** (Figure 5A,B). At the allosteric pocket, both inhibitors **1** and **2** formed halogen interactions with Ala189, Asn193, and Phe196. The strength of interaction with these residues was higher for **1** than **2**, as revealed by the presence of multiple site interactions within an amino acid residue, indicated in the simulation software with red lines (Figure 5D,E). The PTP1B-**3** inhibitor complex at the catalytic pocket showed that **3** fitted well in the active catalytic pocket by forming favorable H-bond interactions with the main interacting amino acid residues Arg221, Cys215, and Lys116 (Figure 5C). In addition, multi-site halogen interactions with Asp181, Cys215, Gln262, Lys120, Ser216, Gln266, Tyr46, and Lys116 were observed, which indicated favorable binding with high affinity and low binding energy (−6.86 kcal/mol).

The PTP1B-**1** inhibitor complex at the catalytic pocket showed four H-bond interactions (Lys116, Asp181, Arg221, and Gln226) at a distance of 2.0 to 2.2 Å. In addition, bromine atoms at C2, C3, and C6 positions were involved in multiple interactions with Thr263, Val184, Gln266, Gly183, Arg221, Phe182, Lys116, and Trp176. Similarly, PTP1B-**2** inhibitor complex displayed three H-bond interactions with Arg221, Gln266, and Gly183 at a distance of 1.9 to 2.1 Å. Three bromine atoms of **2** showed multiple interactions with Arg211, Gln266, Trp179, Thr283, Asp265, Asp181, Lys116, Phe182, and Val184. The catalytic inhibitor, compound A, formed a complex with PTP1B at the catalytic site forming four H-bond interactions with Asp48, Lys116, Lys120, and Asp181 along with other non-polar interactions with Ala217, Tyr46, Met258, and Gln262. Common interacting residues for inhibitors (**1**, **2**, and compound A) at the catalytic pocket involved Asp181 and Lys116, which were determining residues for catalytic inhibition.

### 2.5. Molecular Docking Simulation of α-Glucosidase Inhibition

The 3D structure of α-glucosidase used in biological assays from yeast has not been reported yet. So, isomaltase from *Saccharomyces cerevisiae* co-crystallized with maltose (PDB ID: 3A4A) was used as the α-glucosidase protein because it showed 85% similarity to yeast α-glucosidase (MAL12) through homology modeling [28]. To elucidate the interaction between bromophenols and α-glucosidase (PDB ID: 3A4A), docking simulations were performed using AutoDock 4.2. A summary of binding energies of test compounds and reference ligands, along with a list of amino acid residues involved in H-bond and halogen interactions, are reported in Table 4. Similarly, Figure 4B presents a graphical sketch of inhibitors **1**–**3** at the active site of α-glucosidase, and Figure 6 shows interactions with different amino acid residues at respective binding sites. Bromophenols **1** and **2** displayed allosteric inhibition at allosteric binding site 1 and site 2, respectively.

The optimal conformation of **1** was inside site-1 in the allosteric pocket with the lowest energy of −5.94 kcal/mol, forming H-bond interactions with Thr290 and Glu271 at a distance of 2.2 Å. Halogen interactions were formed with Asp341, Cys342, Ser296, Glu271, Lys16, and Lys13 (Figure 6A). Bromophenol **2** was most appropriately suited to bind in site-2 in the allosteric pocket with the lowest binding energy of −5.61 kcal/mol, forming H-bond interactions with Tyr158, Arg315, Glu411, and Asn415 located at a distance 2.2 Å. Halogen interactions were formed with Arg315, Lys56, Gly160, Phe314, and Leu313 (Figure 6B). The reference allosteric inhibitor (*Z*)-3-butylidenephthalide (Compound D) bound to site-1 of the allosteric pocket with a binding energy of −6.85 kcal/mol, forming two H-bond interactions with Glu296 and His295 along with several non-polar interactions with Trp15, Lys16, Asn259, Ala292, Thr290, Arg294, Leu297, Ser291, Ser298, Asp341, Cys342, and Trp343 at the allosteric site. However, compound C (reference catalytic inhibitor) demonstrated numerous H-bond interactions (Asp69, His112, Tyr158, Gln182, Asp215, Arg213, Ser240, Asp242, Glu277, His280, Asp307, Asp352, and Arg442) at the catalytic pocket along with non-polar interactions involving Tyr72, Lys156, Phe178, Val216, Gln279, Phe303, Arg315, Gln353, His351, and Glu411. Interestingly, **3** showed a mixed mode of inhibition by interacting with both catalytic and allosteric regions (specifically at site 2). Since **3** was a dimer that comprised two bromophenol groups, one of the bromophenol groups bound to the active catalytic pocket and the other bound to the allosteric pocket at site-2 (Figure 4B). Hydroxyl groups at the C-4 and C-5 positions of ring A formed two H-bonds with Ser157 at a distance of 2.1 and 2.3 Å. Halogen interactions between the bromine atom and Asn415, Ser157, Tyr156, and Arg315 were also observed at the catalytic pocket (Figure 6C). However, at site-2 in the allosteric pocket, the -OH groups at C-4′ and C-5′ displayed H-bond interactions with Asp307 and Gln353 located at a distance of 1.9 and 2.2 Å, respectively. Halogen interactions were observed between bromine atoms and His280, Phe303, Gln279, Arg315, Tyr158, and Asp307 at site-2 in the allosteric pocket.

### 2.6. Evaluation of Cytotoxicity in HepG2 Cells

Cytotoxicity of bromophenols on HepG2 cells was evaluated using a 3-(4,5-dimethylthiazol-2-yl)-2,5-diphenyl tetrazolium bromide (MTT) assay. As shown in Figure 7, bromophenols **2** and **3** did not show toxicity until they were used at a concentration of 100 μM. Bromophenol **1** showed a slightly toxic effect at 100 μM. So, considering enzyme inhibition activity and toxicity in HepG2 cells, we performed cell-based experiments at 5, 10, and 20 µM concentrations.

### 2.7. Effects on Glucose Uptake in Insulin-Resistant HepG2 Cells

We measured the effect of **1**–**3** on glucose uptake in insulin-resistant (IR) HepG2 cells. As shown in Figure 8, the glucose uptake was reduced by 36% (to 64%) in the IR group when compared to the normal control group. However, treatment with bromophenols **1**–**3** boosted the insulin-stimulated glucose uptake significantly. Rosiglitazone at a concentration of 10 µM enhanced glucose uptake to 112%, which was comparable to the stimulatory effects of **1** and **2** at 20 µM. Even at a concentration of 10 µM, bromophenol **3** had a greater effect than rosiglitazone and a comparable effect to the normal control group. The pattern of insulin-stimulated glucose uptake in IR-HepG2 cells by bromophenols was similar to the pattern of their PTP1B inhibition.

### 2.8. Effects on PTP1B Expression Level in Insulin-Resistant HepG2 Cells

Increased PTP1B activity attenuates insulin signals and results in insulin resistance, implying that downregulation of PTP1B activity would theoretically enhance insulin sensitivity. In order to determine whether bromophenols **1**–**3** are capable of downregulating PTP1B activity, their effect on PTP1B expression was evaluated using a western blot. As shown in Figure 9, bromophenols reduced the expression of PTP1B in a concentration-dependent manner. Compared to the normal control group, which had low levels of PTP1B expression, the insulin-resistant group demonstrated intensified PTP1B expression levels. When insulin-resistant HepG2 cells were treated with rosiglitazone and/or bromophenols, the expression levels again went down. Interestingly, all tested bromophenols were better at downregulating PTP1B expression than rosiglitazone. Bromophenol **3**, the dimer, once again outperformed other compounds with its excellent ability to reduce PTP1B expression level.

## 3. Discussion

Natural products (NPs) have played a central role in drug discovery and development since 1805 when morphine was isolated in a pure form from opium. To date, the majority of NP-derived drugs have been reported from terrestrial plants, fungi, or microorganisms. However, marine sources have attracted interest because 70% of the earth’s surface is covered by water, and the oceans represent 95% of the biosphere. Furthermore, due to high species diversity in the marine environment, we can also expect a high level of structural diversity in secondary metabolites derived from marine organisms with a high incidence of bioactivity [29,30]. Among diverse groups of chemical entities from marine sources, halogenated secondary metabolites, especially chlorophenolic and bromophenolic metabolites, are most common in red and brown algae. Although chlorine content is higher than bromine in sea water, brominated compounds are more common than chlorinated compounds because bromine is more frequently used by algae to produce organohalogens [9]. With some exceptions, most species of red algae have a relatively high content of bromophenols [31]. Phenol moiety has a tendency to undergo electrophilic bromination. For this reason, bromophenols are the main components of algae responsible for reported anti-diabetic activity, and their potency is attributed to their bromine content and side chains [32]. Anti-diabetic activity of bromophenols has been reported via PTP1B and α-glucosidase inhibition. Despite this, there are limited studies on enzyme inhibition mechanisms, molecular docking simulations, structure–activity relationships, and insulin-sensitizing properties with in vitro cell models. This study addresses all of these topics in order to discover anti-diabetic compounds from *S. latiuscula*.

As part of our continual search for anti-diabetic drugs from marine sources, we discovered three bromophenols from *S. latiuscula*, a red alga. Among different marine algae, red and brown algae have been reported to have anti-diabetic properties via inhibition of carbohydrate hydrolyzing enzymes and PTP1B enzymes, insulin-sensitization, and glucose uptake [33]. However, the particular compounds responsible for inhibiting particular enzymes have not been fully characterized. With an aim to discover and characterize anti-diabetic components from *S. latiuscula*, we performed a bioassay-guided isolation study. Three potent dual inhibitors of PTP1B and α-glucosidase enzymes, namely, 2,3,6-tribromo-4,5-dihydroxybenzyl alcohol (**1**), 2,3,6-tribromo-4,5-dihydroxybenzyl methyl ether (**2**), and bis-(2,3,6-tribromo-4,5-dihydroxybenzyl methyl ether) (**3**), were discovered from the active EtOAc fraction. Studies by Kurihara et al. [24,34] had reported α-glucosidase inhibitory activity for compounds **1** and **3** solely based on IC_50_ values. In their studies, the 50% α-glucosidase enzyme inhibition concentration was 11 and 0.03 μM for **1** and **3**, respectively. In our study, these values were 2.63 and 1.92 μM for **1** and **3**, respectively. This difference can be explained by differences in the experimental conditions, namely, concentrations of enzyme (5 μg/mL vs. 20 μg/mL), concentrations of *p*NPG as substrate (5–20 mM vs. 2.5 mM), incubation times, and purity of tested compounds. No previous study has reported the potential mechanisms of enzyme inhibition via kinetic analysis and molecular docking simulation. The PTP1B inhibitory potential of **2** was reported by Liu et al. [35]. In that report, compound **2** inhibited PTP1B with an IC_50_ value of 3.9 μM. In our study, the 50% inhibitory concentration is approximately twice the previously reported value (8.50 μM). Differences in experimental conditions like enzyme concentrations, type of buffer, and stop solution might be responsible for this difference. There are no other reports of PTP1B inhibition by **1** and **3**.

Our study is the first to explore the mechanism of enzyme inhibition by these bromophenols via enzyme kinetics and molecular docking simulations. Structural differences among the tested bromophenols include a substitution at the C-7 position. In **1**, an -OH group is present at C-7. However, in **2**, the 7-OH group is replaced by a methoxy group. In **3**, another 2,3,6-tribromo-4,5-dihydroxyl methyl ether moiety is linked via O-linkage giving a dimeric form. In enzyme inhibition assays, increasing bromine or bromophenol ring number enhances the PTP1B inhibitory activity as seen for **3**. A similar result could be seen in a recent report by Zhang et al. [36], which showed that an increase in bromophenol group number improved PTP1B inhibitory activity. Activity is similarly affected for α-glucosidase enzyme inhibition. The replacement of 7-OCH_3_ in **2** with the –OH group in **1** enhanced inhibitory activity 2.5-fold. Similarly, the bis-phenol form (**3**) further enhanced the activity 3.8-fold compared to **2**. These variations in extent of α-glucosidase enzyme inhibition might facilitate the discovery and design of more effective anti-diabetic molecules.

In order to have information on the mechanism of enzyme inhibition, we investigated enzyme kinetics and molecular docking. Evaluating enzyme kinetics is an essential component of the drug development process because it enables the monitoring of intermediate molecules during a hit-to-lead optimization process [37]. Results from our kinetic studies revealed that bromophenol **1** and **2** are non-competitive inhibitors of α-glucosidase. Bromophenol **3**, on the other hand is a mixed-type inhibitor (as the inhibitor concentration increases, *K*_m_ increases and *V*_max_ decreases). Bromophenol **1** and **3** are mixed-type inhibitors of PTP1B, and bromophenol **1** is a competitive inhibitor of PTP1B.

Discovery of small molecule PTP1B inhibitors is challenging because the catalytic pocket of PTP1B is relatively shallow, reducing the advantage of a small molecule inhibitor; and the catalytic site is comprised of several polar amino acid residues (e.g., Ser216, Cys215, and Arg221), which selectively bind small molecules with only polar groups [38,39]. In addition, carboxylic acids, phosphonates, sulphonamides, and sulphanamic acids that were initially used in the design of PTP1B inhibitors resulted in poor cell permeability and oral bioavailability. This is why the development of small molecule PTP1B inhibitors using computational and experimental approaches is urgent. In order to predict the binding affinity of ligands to an enzyme’s active site, and to understand the binding mechanism, we performed molecular docking simulations. Molecular docking simulations of ligands **1**–**3** at the active site cavity of PTP1B revealed different modes of inhibition that resulted from H-bond and halogen interactions. Catalytic inhibition was observed in interactions with conserved amino acid residues (Cys215 and Asp181). These amino acid residues, specifically Cys215 in the PTP signature motif (P-loop) and Asp181 in the conserved protein loop (WPD loop), were identified as primary active site residues through structural and kinetic analyses. When an inhibitor/substrate binds to the WPD loop, active sites are blocked, and the phosphocenter of the substrate is dephosphorylated by Cys215 via nucleophilic attack. In addition, electrostatic, hydrophobic, and H-bonding interactions occur at Arg47, Lys120, and Val49 [40]. In addition, Phe182 and Gln262 are important amino acid residues for substrate binding [41]. Similarly, Arg24 and Arg254 are important residues for H-bond interactions with Met258 and Gln262, and van der Waals interactions with Ile219, Asp48, and Val49 at site B, which was reported as a second catalytic site by Puius et al. [42]. Results of PTP1B docking revealed that **1** and **2**, and the reference allosteric inhibitor (compound B), shared common interacting residues Ala189, Asn193, Glu266, Leu192, Phe196, and Phe280 at the allosteric site. These are prime interacting residues at the allosteric site, and **3** did not show interactions with any of these residues. Instead, it showed multiple bonding with prime amino acid residues Asp181, Arg221 and Lys116 that were in common with the reference catalytic inhibitor (compound A). Interestingly, **1** and **2** showed additional interactions with Arg221 and Lys116 at the catalytic site, representing mixed-type inhibition. H-bond interactions of **1**–**3** and reference inhibitors at respective binding sites are similar in number, however, bromine atoms of **1**–**3** showed numerous halogen interactions, which aids in enhancing stability at the binding sites.

Docking simulations for α-glucosidase revealed the involvement of two sites (site-1 and site-2) in the allosteric pocket. The docking simulation was conducted with 15 independent genetic algorithms (GAs) with the default parameters. In case of **1**, all fifteen complexes were docked into the site-1 allosteric pocket, like (Z)-3-butylidenephthalide (compound D), a reference allosteric inhibitor. In the case of **2**, all fifteen complexes were docked into the site-2 allosteric pocket. However, in the case of **3**, seven of fifteen enzyme-**3** complexes were docked into the site-2 allosteric pocket, while the remaining eight complexes were docked into both the catalytic pocket and the site-2 allosteric pocket with the lowest binding energy. Therefore, we concluded that **3** is a mixed-type inhibitor that binds to both the catalytic and site-2 of the allosteric pocket. Ding et al. [43] predicted five binding cavities on α-glucosidase, one of which is an active pocket and the other four of which are allosteric pockets away from the active site that bind non-competitive inhibitors. Similarly, Didem et al. [44] had reported details of the sitemap results for α-glucosidase enzyme from *Saccharomyces cerevisiae*, including binding residues at the catalytic site and four allosteric binding sites (ABS1-ABS4). In our docking result, allosteric inhibition involved only two different sites, which we named as site-1 and site-2. Bromophenol **1** showed allosteric inhibition by binding to site-1 of the allosteric pocket via various H-bond and other interacting residues (Table 4), of which Glu271 and Ala292 are common residues at ABS4. Similarly, **2** bound to a different site other than site-1, and we named it site-2. Interacting residues at site-2 that were involved in binding of **2** were Tyr158, Asn415, Lys156, Gly160, and Phe314. These residues were reported to interact at the ABS1 allosteric site. The reference allosteric inhibitor (Z)-3-butylidenephthalide (compound D) and **1** demonstrated the interaction with common residues Thr290, Lys16, Ala292, Cys342, Ser298, Leu297, and Trp343, demonstrating that these two bound at the same site (site-1). These results suggest that **1** and **2** are non-competitive inhibitors that bind to site-1 and site-2 in the allosteric pocket. Acarbose (compound C) is a reference catalytic inhibitor that bound to the catalytic pocket with the lowest binding energy and demonstrated the involvement of H-bond interacting residues Asp69, His112, Gln182, Asp215, Arg213, Ser240, Asp242, Glu277, His280, Asp307, Asp352, and Arg442, and non-polar interactions with Tyr72, Lys156, Phe178, Val216, Gln279, Phe303, Gln353, His351, and Glu411. Interestingly, **3** showed interactions with Asp307, His280, Asp352, Gln353, Gln279, and Phe303 at the catalytic site similar to acarbose, and with Tyr158, Asn415, and Arg315 common to **2** at site-2 of allosteric pocket. These results also indicate that **3** is a mixed mode inhibitor that binds one bromophenol ring to the active pocket and the other bromophenol ring to site-2 in the allosteric pocket (Figure 6C). In addition, halogen interactions between bromine and Tyr158 and Arg315 further stabilize the binding of **3** to both binding pockets. The overall results of our docking simulation and enzyme kinetics analysis confirm that the allosteric inhibition observed for this enzyme is due to non-competitive binding. This is in line with research that reports that allosteric inhibitors generally function through non-competitive or mixed binding [45]. It is notable that compounds with more bromine atoms and/or bromophenol rings have increased activity, which is likely due to the involvement of a large number of halogen interactions that play a vital role in the positioning of inhibitors into the active site with a high degree of stability.

Dysregulated insulin signaling is a main part of T2DM, and insulin mediates metabolic and mitogenic effects when it binds to insulin receptors [46]. PTP1B plays a critical role in switching off insulin signaling by phosphorylating and inactivating the insulin receptor. Specifically, PTP1B interacts with insulin receptors and insulin-receptor substrate-1 to hydrolyze phosphorylated tyrosine (which is induced by insulin), leading to impaired glucose uptake [47]. Increased expression of PTP1B in insulin-sensitive tissues is a marker of insulin resistance, while decreased expression enhances insulin sensitivity, thereby improving glucose uptake and insulin signaling [48]. We developed insulin-resistant HepG2 cells in order to evaluate the insulin sensitizing and glucose uptake potential of bromophenols via PTP1B downregulation. Our western blot results suggested that PTP1B was highly expressed in the insulin-resistant group while it was weakly expressed in the normal control group. Treatment with different concentrations of bromophenols downregulated the expression of PTP1B in a concentration-dependent manner. The results of both the 2-NBDG glucose uptake assay and PTP1B expression analysis support each other. In summary, the anti-diabetic potential of bromophenols **1**–**3** is attributed to the inhibition of both PTP1B and α-glucosidase, which enhances insulin-sensitivity and glucose uptake through the downregulation of PTP1B in insulin-sensitive tissues. Describing a detailed molecular mechanism is a critical next step in understanding this process. Furthermore, in vivo study is of the utmost importance in validating our findings.

## 4. Materials and Methods

### 4.1. Chemicals and Reagents

Three test compounds were isolated from EtOAc fraction of methanol extract of *S. latiuscula*: 2,3,6-tribromo-4,5-dihydroxybenzyl alcohol, 2,3,6-tribromo-4,5-dihydroxybenzyl methyl ether, and bis-(2,3,6-tribromo-4,5-dihydroxybenzyl methyl ether). Acarbose, ethylenediaminetetraacetic acid (EDTA), *p*-nitrophenyl α-D-glucopyranoside (*p*NPG), *p*-nitrophenyl phosphate (*p*NPP), rosiglitazone, dimethyl sulfoxide (DMSO), and yeast α-glucosidase were purchased from Sigma-Aldrich Co. (St. Louis, MO, USA). Human recombinant protein tyrosine phosphatase (PTP1B) was purchased from Biomol International LP (Plymouth Meeting, PA, USA). Fetal bovine serum (FBS), minimum essential medium (MEM), sodium pyruvate, penicillin-streptomycin, and nonessential amino acids were purchased from Gibco-BRL Life Technologies (Grand Island, NY, USA). The fluorescent D-glucose analogue and 2-NBDG was purchased from Life Technologies (Carlsbad, CA, USA). Human insulin was purchased from Eli Lilly (Fegersheim, France). All other chemicals and solvents used were of reagent grade and acquired from commercial sources.

### 4.2. Algal Material

Leafy thalli of *S. latiuscula* (Harvey) Yamada were collected from Cheongsapo, Busan, Korea, in January 2016. This red alga was authenticated by an algologist, Dr. K. W. Nam at the Department of Marine Biology, Pukyong National University. A voucher specimen (No. 20160140) has been deposited in the laboratory of Prof. J. S. Choi, Pukyong National University.

### 4.3. Extraction and Fractionation

The clean and dried leafy thali of *S. latiuscula* (Harvey) Yamada (700 g) was extracted with MeOH (5 L × 3 times) successively for 3 h at a time under reflux. The extract was filtered and concentrated until dry in vacuo at 40 °C to obtain dry MeOH extract (207.38 g). The resulting MeOH extract was successively partitioned with different solvent soluble fractions including a CH_2_Cl_2_ fraction (43.33 g), EtOAc fraction (9.47 g), *n*-BuOH fraction (21.89 g), and H_2_O fraction (118.96 g).

### 4.4. Isolation of Bromophenol Derivatives from Symphyocladia latiuscula

The EtOAC fraction was passed through sephadex LH-20 gel using CH_2_Cl_2_–MeOH in a one-to-one ratio to obtain 20 subfractions (E1-E20). Subfraction E2 (172 mg) was chromatographed on a Si gel column with CH_2_Cl_2_–MeOH–H_2_O in the ratio of 7:1:0.1 to get 10 subfractions (E2.1–E2.10). Among ten subfractions, E2.7 was purified on an RP-18 gel column employing 75% acetonitrile as an eluent to obtain **1** (11.7 mg). In addition, subfraction E7 (258 mg) was subjected to Si gel column chromatography using CH_2_Cl_2_–MeOH–H_2_O in the ratio of 4:1:0.1 to get 10 subfractions (E7.1–E7.10). Among ten subfractions, E7.4 was purified on an RP-18 gel column employing 40% acetonitrile as an eluent to obtain **2** (9.4 mg). Similarly, subfraction E11 (380 mg) was subjected to Si gel column chromatography using EtOAc–MeOH–H_2_O in the ratio of 24:3:2 to get 15 subfractions (E11.1–E11.15). Among those subfractions, E11.9 was purified on an RP-18 gel column employing 40% acetonitrile as an eluent to obtain **3** (8.0 mg). The chemical structures of isolated compounds (Figure 1) were characterized referring to published spectral data (1D/2D NMR and ESIMS) [18,22,24,27]. Purity of compounds was above 98%, as evidenced by spectral data.

2,3,6-Tribromo-4,5-dihydroxybenzyl alcohol (**1**): Obtained as pale yellow solid. ^1^H NMR (CD_3_OD, 600 MHz) δ 4.95 (2H, s, H-7); ^13^C-NMR (CD_3_OD, 150 MHz) δ 147.42 (C-5), 145.05 (C-4), 128.65 (C-1), 119.34 (C-6), 114.68 (C-3), 113.96 (C-2), and 74.41 (C-7). HRFAB MS *m*/*z* 373.7743 (calculated for C_7_H_5_Br_3_O_3_ 373.7789).

2,3,6-Tribromo-4,5-dihydroxybenzyl methyl ether (**2**): Obtained as amorphous light brown powder; ^1^H NMR (CD_3_OD, 600 MHz) δ 4.89 (2H, s, H-7), 3.39 (3H, s, H-8); ^13^C-NMR (CD_3_OD, 150 MHz) δ 146.87 (C-5), 144.89 (C-4), 129.26 (C-1), 119.12 (C-6), 114.59 (C-3), 114.12 (C-2), 76.52 (C-7), and 58.39 (CH_3_, H-8). HRFAB MS *m*/*z* 387.7904 (calculated for C_8_H_7_Br_3_O_3_ 387.7945).

Bis-(2,3,6-tribromo-4,5-dihydroxybenzyl methyl ether) (**3**): Obtained as amorphous light brown powder; ^1^H NMR (CD_3_OD, 600 MHz) δ 4.95 (4H, s, H-7 and H-7′); ^13^C-NMR (CD_3_OD, 150 MHz) δ 146.79 (C-5 and C-5′), 144.69 (C-4 and C-4′), 129.22 (C-1 and C-1′), 119.59 (C-6 and C-6′), 115.03 (C-3 and C-3′), 114.09 (C-2 and C-2′), and 74.44 (C-7 and C-7′). HRESIMS [M + Na]^+^
*m*/*z* 752.5364 calculated for C_14_H_8_^79^Br_6_O_5_, *m*/*z* 754.5344 calculated for C_14_H_8_^79^Br_5_^81^BrO_5_, *m*/*z* 756.5324 calculated for C_14_H_8_^79^Br_4_^81^Br_2_O_5_, *m*/*z* 758.5304 calculated for C_14_H_8_^79^Br_3_^81^Br_3_O_5_, *m/z* 760.5284 calculated for C_14_H_8_^79^Br_2_^81^Br_4_O_5_, *m*/*z* 762.5266 calculated for C_14_H_8_^79^Br^81^Br_5_O_5_, and *m*/*z* 764.5250 calculated for C_14_H_8_^81^Br_6_O_5._

### 4.5. Protein Tyrosine Phosphate 1B (PTP1B) Inhibitory Assay

The potential for MeOH extract, solvent-soluble fractions, and isolated compounds to inhibit PTP1B enzyme activity was evaluated using the *p*NPP substrate [49]. Experimental conditions and laboratory procedures were similar to those published in our previous report [50]. Ursolic acid was used as a reference compound and results are presented as IC_50_ values (mean ± standard deviation) from triplicate experiments.

### 4.6. α-Glucosidase Inhibitory Assay

The α-glucosidase inhibition assay procedure, previously reported by Li et al. [51], was followed to measure the release of *p*NPG at 405 nm using a microplate spectrophotometer (Molecular Devices). Acarbose was used as a reference compound.

### 4.7. Kinetic Study Against PTP1B and α-Glucosidase

The modes of enzyme inhibition were evaluated by monitoring the effects of different concentrations of the substrates (1, 2, or 4 mM *p*NPP for PTP1B and 1.25, 2.5, or 5 mM *p*NPG for α-glucosidase) on Dixon plots (single reciprocal plots). Lineweaver–Burk plots for inhibition of PTP1B were obtained in the presence of various concentrations of the test compounds (0, 5, 8, and 10 μM for **1**; 0, 2.5, 9, and 18 μM for **2**; and 0, 2.5, 5, and 7 μM for **3**). Similarly, for the inhibition of α-glucosidase, the test concentrations used were 0, 2, 3, and 5 μM for **1**; 0, 1.5, 7.5, and 15 μM for **2**; and 0, 0.125, 0.5, and 2 μM for **3**. The enzymatic procedure consisted of the same aforementioned assay method. The inhibition constants (*K*_i_) were determined by interpreting Dixon plots, where the value of the *x*-intercept indicated the value of −*K*_i_ [52,53].

### 4.8. Molecular Docking Simulation of PTP1B and α-Glucosidase Inhibition

The complex structure of PTP1B with its selective allosteric inhibitor 3-(3,5-dibromo-4-hydroxy-benzoyl)-2-ethyl-benzofuran-6-sulfonic acid (4-sulfamoyl-phenyl)-amide (known as ‘compound B’ in our study) (PDB ID: 1T49) and the 3D structure of the selective catalytic inhibitor 3-({5-[(*N*-acetyl-3-{4-[(carboxycarbonyl)(2-carboxyphenyl)amino]-1-naphthyl}-L-alanyl)-amino]pentyl}oxy)-2-naphthoic acid (known as ‘compound A’ in our study) were obtained from the Research Collaboratory for Structural Bioinformatics (RCSB) Protein Data Bank website [54] and PubChem Compound (NCBI) with a compound CID of 447410. The structure of α-glucosidase with its catalytic ligand α-D-glucose (PDB ID: 3A4A) and the structures of acarbose (known as ‘Compound C’ in our study) and (*Z*)-3-butylidenephthalide (BIP) (known as ‘Compound D’ in our study) were obtained from the RCSB Protein Data Bank website [55] and PubChem Compound (NCBI) with compound CIDs of 41774 and 5352899, respectively. Protein preparation was conducted using Accelrys Discovery Studio 17.2 (Accelrys, Inc., San Diego, CA, USA). The binding areas of compound A, compound B, compound C, and compound D were considered to be the most convenient regions for ligand binding in the docking simulation. The 3D structures of compounds **1**–**3** were constructed using Chem3D Pro v12.0 with pH adjusted to 7 using MarvinSketch (ChemAxon, Budapest, Hungary). A Lamarckian genetic algorithm (GA) method was used for docking simulations. Gasteiger charges were added by default, the rotatable bonds were set with AutoDockTools (ADT), and all torsions were allowed to rotate. The docking simulation was conducted with 15 independent GAs with the default parameters. Results were analyzed and visualized using University of California, San Francisco (UCSF)’s Chimera tool (http://www.cgl.ucsf.edu/chimera/).

### 4.9. Cell Culture, 3-(4,5-dimethylthiazol-2-yl)-2,5-diphenyl tetrazolium bromide (MTT) Assay, and Insulin Resistance Induction

HepG2 (human hepatocarcinoma) cells obtained from the American Type Culture Collection (HB-8065; Manassas, VA, USA) were maintained in 10% FBS MEM at 37 °C in a humidified atmosphere with 5% CO_2_. Cytotoxicity was evaluated using the MTT assay [56]. Following previously described methods [57], an insulin-resistant HepG2 cell model was established. All other experimental conditions and procedures were similar to those reported in our previous paper [58].

### 4.10. Glucose Uptake Assay

The relative percent glucose uptake upon sample treatment in IR-HepG2 cells was evaluated by a flourimetric method using the fluorescent D-glucose analog 2-NBDG and measured on a fluorescence microplate reader (FL×800, Bio-Tek Instruments, Inc., Winooski, VT, USA) at 485 nm excitation and 528 nm emission wavelengths. Detailed experimental conditions and procedures are reported in our previous publication [59].

### 4.11. Preparation of Cell Lysates and Western Blot Analysis

Insulin-resistant HepG2 cells were exposed to a range of concentrations of **1**–**3** or rosiglitazone (10 μM) in 24-well plates for 24 h followed by stimulation with 100 nM insulin diluted in SFMEM for 30 min. Cells were then washed twice with ice cold PBS, collected, and lysed with sample buffer. Protein was quantified by the modified Bradford protein assay kit using bovine serum albumin (BSA) as a standard, electrophoresed on sodium dodecyl sulfate-polyacrylamide gels (Bio-Rad, Hercules, CA, USA), and then transferred to polyvinylidene difluoride (PVDF) membranes (Immobilon-P; Millipore, Burlington, MA, USA) using the Pierce G2 Fast Blotter (Thermo Fisher Scientific Inc, USA) at 100 V for 60 min in a standard semi-dry system. Membranes were blocked in blocking buffer and incubated overnight with primary antibodies at 4 °C on a shaker, followed by incubation with the appropriate secondary antibodies for 2 h at room temperature. The membranes were washed three times with tris-buffered saline with tween (TBST) and the protein bands were visualized with the Supersignal West Pico Chemiluminescence Substrate (Pierce, Rockford, IL, USA) on X-ray films (Kodak, Rochester, NY, USA) and quantified using CS analyzer software (Atto Corp., Tokyo, Japan). Detailed laboratory procedures and sample buffer and antibody solution compositions are reported in our previous paper [59].

### 4.12. Statistical Analysis

One-way analysis of variance (ANOVA) and Student’s *t*-test (Systat Inc., Evanston, IL, USA) were used to determine the statistical significance. Values of *p* < 0.05, 0.01, and 0.001 were considered significant. All results are presented as the mean ± standard deviation (SD) of triplicate experiments.

## 5. Conclusions

In summary, bioassay-guided isolation led to the discovery of three potent dual PTP1B and α-glucosidase inhibitors from a red alga, *S. latiuscula* (Harvey) Yamada: 2,3,6-tribromo-4,5-dihydroxybenzyl alcohol (**1**), 2,3,6-tribromo-4,5-dihydroxybenzyl methyl ether (**2**), and bis-(2,3,6-tribromo-4,5-dihydroxybenzyl methyl ether) (**3**). Notably, these bromophenols enhanced insulin sensitivity in insulin-resistant HepG2 cells and aided glucose uptake by downregulating PTP1B expression. Structural insights into the PTP1B and α-glucosidase enzymes inhibition revealed the importance of bromine and phenolic group number. In addition, molecular docking simulation revealed the mechanisms by which H-bonds and other interactions between the active ligands and enzymes are critical for inhibition. Bromophenols from *S. latiuscula* may represent a novel class of anti-diabetic drugs. However, in vivo study is of the utmost importance in confirming these findings, and further studies on the molecular mechanisms of insulin signaling in HepG2 cells is underway.

## Figures and Tables

**Figure 1 marinedrugs-17-00166-f001:**
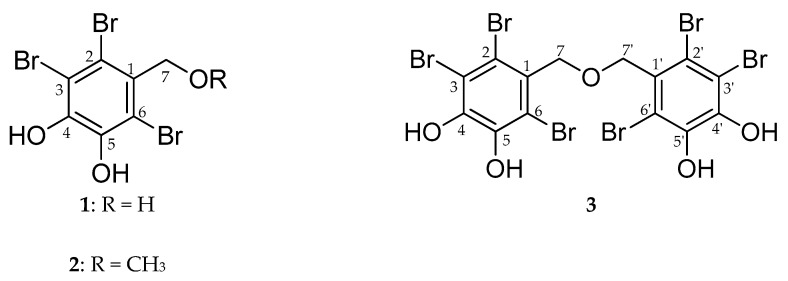
Structure of the compounds isolated from the EtOAc fraction of *S. latiuscula*.

**Figure 2 marinedrugs-17-00166-f002:**
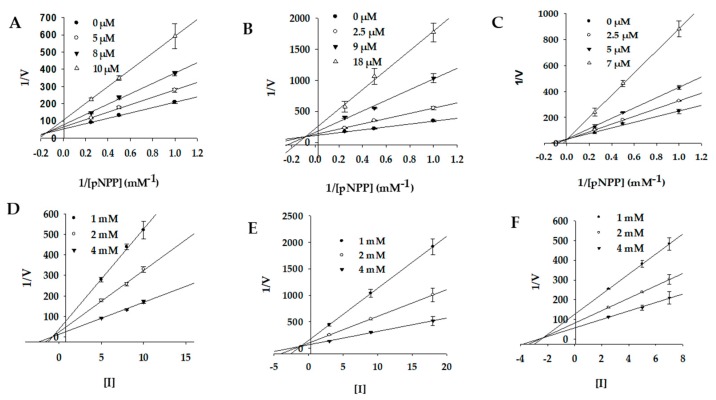
Enzyme kinetic plots of PTP1B inhibition by bromophenols **1**–**3**, respectively. (**A**–**C**) Lineweaver–Burk plots, and (**D**–**F**) Dixon plots.

**Figure 3 marinedrugs-17-00166-f003:**
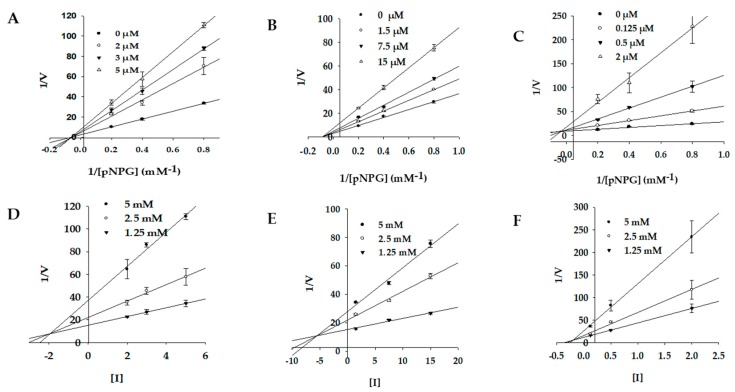
Enzyme kinetic plots of α-glucosidase inhibition by isolated compounds **1**–**3**, respectively. (**A**–**C**) Lineweaver-Burk plots, and (**D**–**F**) Dixon plots.

**Figure 4 marinedrugs-17-00166-f004:**
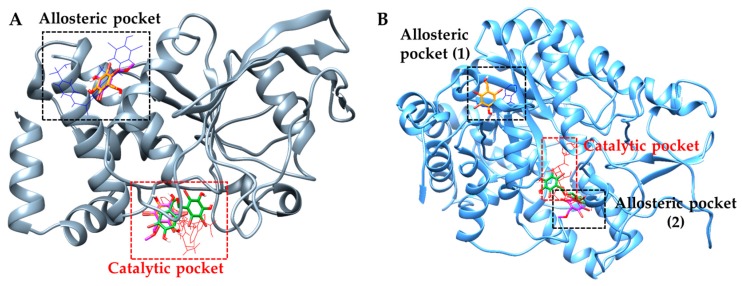
Molecular docking results of bromo-compounds from *S. latiuscula* in the PTP1B (**A**) and α-glucosidase (**B**) along with positive controls. The chemical structures of compounds **1**, **2**, and **3** are shown in orange, purple, and green colored sticks, respectively. Catalytic and allosteric standard compounds are indicated by red and black frames, respectively.

**Figure 5 marinedrugs-17-00166-f005:**
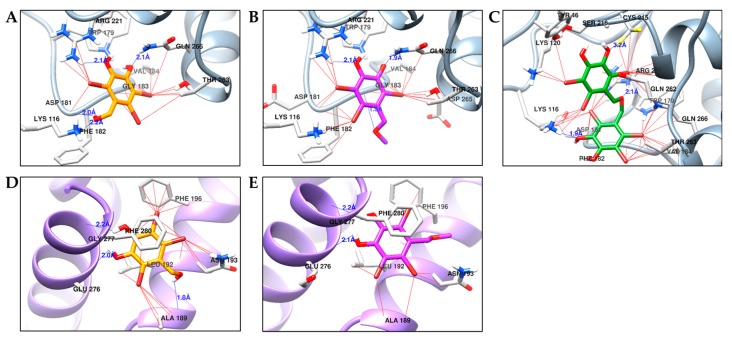
Molecular docking results of bromo-compounds in the catalytic ((**A**) for **1**, (**B**) for **2**, and (**C**) for **3**) and allosteric sites ((**D**) for **1** and (**E**) for **2**) of PTP1B enzyme (1T49). The chemical structures of compounds **1**, **2**, and **3** are shown in orange, purple, and green colored sticks, respectively. H-bond and halogen bond between bromine of **1**–**3** and enzyme residues are indicated by blue and red lines, respectively.

**Figure 6 marinedrugs-17-00166-f006:**
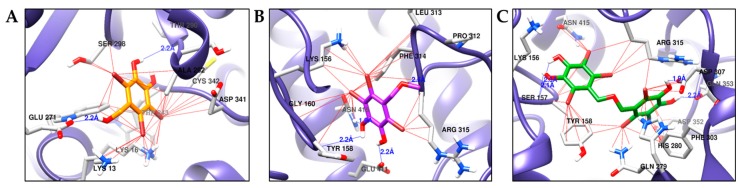
Molecular docking results of bromo-compounds (**1**–**3**) from *S. latiuscula* in the allosteric sites (**A**–**C**) of α-glucosidase enzyme (3A4A). The chemical structures of compounds **1**, **2**, and **3** are shown in orange, purple, and green colored sticks, respectively. H-bond and halogen-bond between bromine of **1**–**3** and enzyme residues are indicated by blue and red lines, respectively.

**Figure 7 marinedrugs-17-00166-f007:**
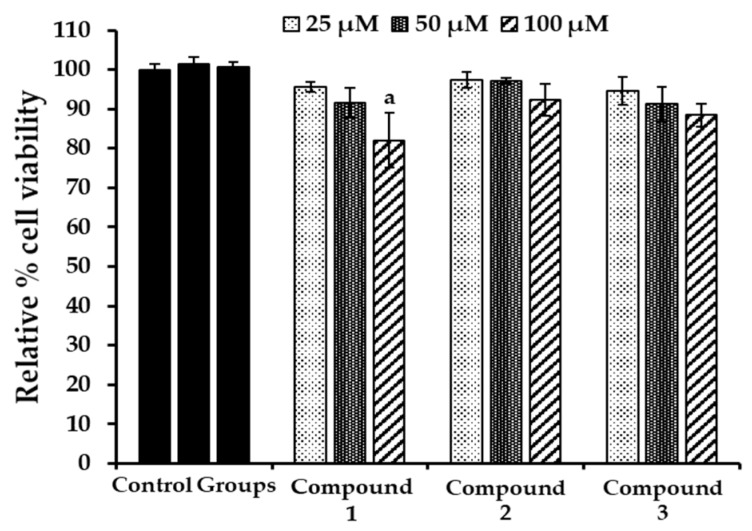
Effect of bromophenols **1**–**3** on cell viability in HepG2 cells. Cell viability was determined using the 3-(4,5-dimethylthiazol-2-yl)-2,5-diphenyl tetrazolium bromide (MTT) method. Cells were pretreated with the indicated concentrations (25, 50, and 100 μM) of test compounds for 24 h. Data shown represent mean ± standard deviation of triplicate experiments. ^a^
*p* < 0.05 indicates significant differences from the control group.

**Figure 8 marinedrugs-17-00166-f008:**
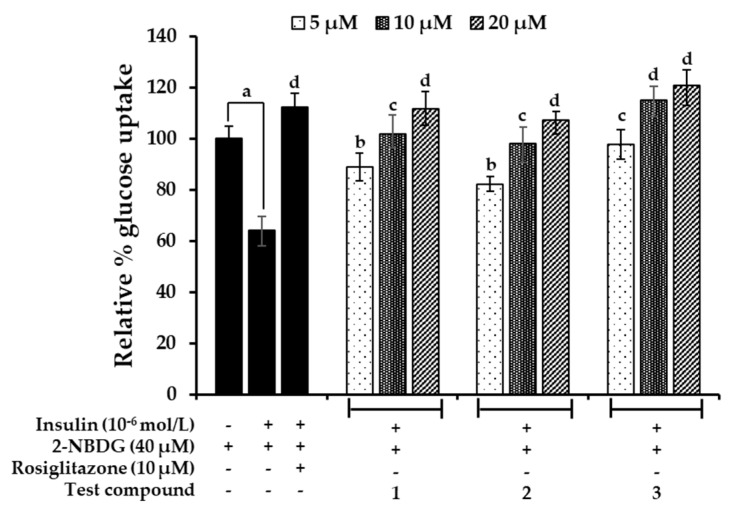
Effect of bromophenols **1**–**3** on insulin-stimulated glucose uptake in insulin-resistant HepG2 cells. The glucose uptake assay was performed using the fluorescent D-glucose analogue 2-NBDG and 10^−6^ mol/L insulin was to induce insulin resistance. Insulin-resistant HepG2 cells were treated with the indicated concentrations of test compounds or rosiglitazone for 24 h, and insulin-stimulated 2-NBDG uptake was measured. Values represent the mean ± standard deviation of triplicate experiments. ^a^
*p* < 0.01 indicates significant differences from the control group; ^b^
*p* < 0.05, ^c^
*p* < 0.01, and ^d^
*p* < 0.001 indicate significant differences from the insulin-resistant group.

**Figure 9 marinedrugs-17-00166-f009:**
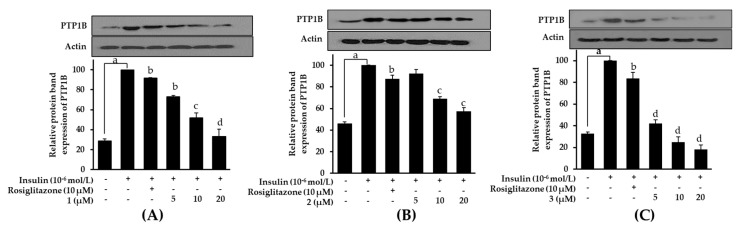
Effects of bromophenol **1** (**A**), **2** (**B**), and **3** (**C**) on protein tyrosine phosphatase 1B (PTP1B) expression levels in insulin-resistant HepG2 cells. Western blotting was performed and protein band intensities were quantified by densitometric analysis. Upper panels display representative blots. Equal protein loading was ensured and normalized against β-actin levels. Values represent the mean ± standard deviation of three independent experiments; ^a^
*p* < 0.001 indicates significant differences from the control group; ^b^
*p* < 0.05, ^c^
*p* < 0.01, and ^d^
*p* < 0.001 indicate significant differences from the insulin-resistant group.

**Table 1 marinedrugs-17-00166-t001:** Tyrosine phosphatase 1B (PTP1B) and α-glucosidase inhibitory activities of MeOH extract and different solvent-soluble fractions of *Symphyocladia latiuscula.*

Test Samples	Yield (%) ^a^	IC_50_ Values (µg/mL) ^b^ ± SD	
		PTP 1B	α-Glucosidase
MeOH	29.62 *	9.01 ± 0.33 ^e^	91.58 ± 9.66 ^f^
CH_2_Cl_2_	20.89	6.95 ± 0.94 ^f^	16.76 ± 1.90 ^h^
EtOAc	4.56	2.79 ± 0.11 ^g^	6.71 ± 0.15 ^i^
*n*-BuOH	10.55	8.72 ± 0.39 ^e^	26.15 ± 0.14 ^g^
H_2_O	57.36	55.03 ± 1.35 ^d^	867.70 ± 42.20 ^d^
Acarbose ^c^	-	-	121.33 ± 2.24 ^e^
Ursolic acid ^c^	-	6.53 ± 0.23 ^f^	-

^a^ Yield (%): The yield (*w*/*w*) percentage of each fraction of the MeOH extract. ^b^ The 50% inhibition concentrations (IC_50_, µg/mL), expressed as the mean ± SD of triplicate experiments. ^c^ Positive controls. ^d–i^ Means with different letters are significantly different on Duncan’s test at *p* < 0.05. * The % yield calculated on dry alga material.

**Table 2 marinedrugs-17-00166-t002:** PTP1B and α-glucosidase inhibitory potential of bromo-compounds from *S. latiuscula*.

Compounds	PTP1B (*n* = 3)			α-Glucosidase (*n* = 3)		
	IC_50_ (µM) ^a^	Inhibition Type ^b^	*K*_i_ (µM) ^c^	IC_50_ (µM) ^a^	Inhibition Type ^b^	*K*_i_ (µM) ^c^
**1**	7.74 ± 0.14 ^f^	Mixed-type	1.19	2.63 ± 0.11 ^g^	Noncompetitive	1.92
**2**	8.50 ± 0.45 ^e^	Mixed-type	2.40	7.24 ± 0.02 ^f^	Noncompetitive	5.54
**3**	5.29 ± 0.08 ^g^	Competitive	2.25	1.92 ± 0.02 ^h^	Mixed-type	0.21
Acarbose ^d^	-	-	-	212.66 ± 0.35 ^e^	-	-
Ursolic acid ^d^	8.66 ± 0.82 ^e^	-	-	-	-	-

^a^ The 50% inhibitory concentration (µM) expressed as mean ± SD of triplicate experiments. ^b^ Inhibition type determined from Lineweaver–Burk plots. ^c^ The inhibition constant (*K*_i_) was determined from Dixon plots. ^d^ Used as reference drugs. ^e–h^ Means with different letters are significantly different with Duncan’s test at *p* < 0.05.

**Table 3 marinedrugs-17-00166-t003:** Binding energy and interaction residues of bromo-compounds from *S. latiuscula* against PTP1B (1T49).

Compounds	Binding Energy (kcal/mol)	H-bond Interactions	Other Interactions
**1**	−5.96	Lys116, Asp181, Arg221, Gln226	Thr263, Val184, Gln266, Gly183, Arg221, Phe182, Lys116, Trp179
	−5.83	Ala189, Glu276, Gly277	Ala189, Leu192, Asn193, Phe196, Phe280, Glu276, Gly277
**2**	−5.80	Arg211, Gln266, Gly183	Arg211, Gln266, Trp179, Thr283, Asp265, Asp181, Lys116, Phe182, Val184
	−5.83	Glu276, Gly277	Ala189, Leu192, Asn193, Phe280, Glu276, Gly277, Phe196
**3**	−6.86	Lys116, Cys215, Arg221	Tyr46, Lys120, Lys116, Asp181, Phe182, Val184, Thr263, Gln266, Trp179, Gln262, Arg221, Cys215, Ser216
Compound A ^a, b^	−7.78	Asp48, Lys116, Lys120, Asp181	Ala217, Tyr46, Met258, Gln262, Ala217
Compound B ^a, c^	−11.30	Glu276, Lys279, Phe280, Asn193	Leu192, Phe196, Ile281, Phe280, Ala189, Lys197

^a^ Compound A, 3-({5-[(*N*-acetyl-3-{4-[(carboxycarbonyl)(2-carboxyphenyl)amino]-1-naphthyl}-L-alanyl)amino]pentyl}oxy)-2-naphthoic acid; compound B, 3-(3,5-dibromo-4-hydroxy-benzoyl)-2-ethyl-benzo-furan-6-sulfonic acid (4-sulfamoyl-phenyl)-amide. ^b^ Reported catalytic inhibitor. ^c^ Reported allosteric inhibitor.

**Table 4 marinedrugs-17-00166-t004:** Binding energy and interaction residues of bromo-compounds from *S. latiuscula* against α-glucosidase (3A4A).

Compounds ^a^	Binding Energy (kcal/mol)	H-Bond Interactions	Other Interactions
**1**	−5.94	Thr290, Glu271	Lys13, Lys16, Trp343, Cys342, Ala292, Thr290, Leu297, Ser298, Glu271
**2**	−5.61	Tyr158, Arg315, Glu411, Asn415	Lys156, Tyr158, Gly160, Pro312, Leu313, Phe314, Arg315, Glu411, Asn415
**3**	−8.06	Ser157, Asp307, Gln353	Tyr158, Lys156, Ser157, Asn415, Arg315, Asp307, Gln353, Phe303, Asp352, His280, Gln279
Compound C ^b^	−8.60	Asp69, His112, Tyr158, Gln182, Asp215, Arg213, Ser240, Asp242, Glu277, His280, Asp307, Asp352, Arg442	Tyr72, Lys156, Phe178, Val216, Gln279, Phe303, Arg315, Gln353, His351, Glu411
Compound D ^c^	−6.85	Glu296, His295	Trp15, Lys16, Asn259, Ala292, Thr290, Arg294, Leu297, Ser291, Ser298, Asp341, Cys342, Trp343

^a^ Compound C, acarbose; compound D, (*Z*)-3-butylidenephthalide. ^b^ Reported catalytic inhibitor. ^c^ Reported allosteric inhibitor.

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
