# Peer review of "Anti-Diabetic Activity of 2,3,6-Tribromo-4,5-Dihydroxybenzyl Derivatives from Symphyocladia latiuscula through PTP1B Downregulation and α-Glucosidase Inhibition"

_marinedrugs, 2019, doi:10.3390/md17030166_

Round 1

Reviewer 1 Report

The authors  have reported the antidiabetic properties of three bromophenols isolated from the red alga Symphyocladia latiuscula.  Considering the impact of Diabetes on public health, this topic is of great interest and every progress utilizable in therapeutic field is of great importance. The authors focus this research on PTP1B and α-glucosidase, two key enzymes involved in T2DM pathologies. Moreover the authors report  a large amount of data to support their  preliminary study on PTP1B and α-glucosidase inhibitory activities, by means of enzyme kinetics, molecular docking simulations, and glucose uptake in IR HepG2 cells.

In my opinion, this is an excellent work and it can be accepted for publication in Marine Drugs after the following minor revisions (I will list some points below):

1.         Title: please use lowercase letter for “Through”

2.       Line 20: please remove enzymes in this sentence

3.       Line 82: modify the sentence as "…in silico molecular modelling on the enzymes used in inhibion assay"

4.       Line 88 type Symphyocladia latiuscula

5.       Line 89-90: please modify in: francions obtained by MeOH extract partition with different solvents.

6.       Line 94: please modify in: of both the enzymes with more promising resuls….

7.       Line 95: please change promising with strong

8.       Line 96: please modify in:  Because this latter fraction gave the most promising inhibition…

9.       Line 101: the author should distinguish between the MeOH % yield (calculated on dry alga material) and the fraction % yield (calculated on MeOH extract).

10.   In the paragraph 2.2. the authors should report a short period about the purification of EtOAc fraction to yield the pure compounds 1-3.

11.   Line 117: please type IC50 value instead of 50% inhibitory concentration…

12.   Line 125: please erase enzyme

13.   Line 150: please change blue lines with black frames

14.   Line 205: please specify that site-1 is in the allosteric pocket (the same for site-2)

15.   The Section 3 should dedicated to the discussion of results as done, however the authors introduce again the topic of the research in an extensive way. Please revise this part. According to me the paragraph from line 292 to 313 can be removed and if necessary integrate it in the introduction, without repeating the concepts.

16.   I think the differences between the data presented in the manuscript for α-glucosidase (lines 326-329) and those previously reported can be  related with the enzyme and substrate concentrations, but not with the base concentration. Another variable could be the different purity of tested compounds.

17.   Line 340: please change bis form with “a dimeric form”

18.   Line 348: please modify the sentence as: “In order to have information on the mechanism …”

19.   Line 352: the sentence “In non-competitive enzyme inhibition Kic = Kiu” makes no sense in this form, it can be omitted.

Author Response

Authors are thankful to Reviewer 1 for constructive comments and suggestions. We have revised our manuscript according to the suggestions and point-to-point responses to reviewer’s comments (in italics) are mentioned below.

Reviewer 1

The authors have reported the antidiabetic properties of three bromophenols isolated from the red alga Symphyocladia latiuscula. Considering the impact of Diabetes on public health, this topic is of great interest and every progress utilizable in therapeutic field is of great importance. The authors focus this research on PTP1B and α-glucosidase, two key enzymes involved in T2DM pathologies. Moreover, the authors report a large amount of data to support their preliminary study on PTP1B and α-glucosidase inhibitory activities, by means of enzyme kinetics, molecular docking simulations, and glucose uptake in IR HepG2 cells.

In my opinion, this is an excellent work and it can be accepted for publication in Marine Drugs after the following minor revisions (I will list some points below):

1. Title: please use lowercase letter for “Through”

Reply: We have corrected this in revised version of manuscript.

2. Line 20: please remove enzymes in this sentence

Reply: Removed

3. Line 82: modify the sentence as "…in silico molecular modelling on the enzymes used in inhibition assay"

Reply: Modified

4. Line 88 type Symphyocladia latiuscula

Reply: Abbreviation has been replaced with full name as suggested.

5. Line 89-90: please modify in: fractions obtained by MeOH extract partition with different solvents.

Reply: Modified

6. Line 94: please modify in: of both the enzymes with more promising results….

Reply: Modified

7. Line 95: please change promising with strong

Reply: The word ‘promising’ has been replaced with ‘strong’.

8. Line 96: please modify in:  Because this latter fraction gave the most promising inhibition…

Reply: Modified

9. Line 101: the author should distinguish between the MeOH % yield (calculated on dry alga material) and the fraction % yield (calculated on MeOH extract).

Reply: Information on footnote has been added as “* The % yield calculated on dry alga material” in the revised version.

10. In the paragraph 2.2. the authors should report a short period about the purification of EtOAc fraction to yield the pure compounds 1-3.

Reply: A brief information has been included in a revised version.

11. Line 117: please type IC50 value instead of 50% inhibitory concentration…

Reply: Corrected

12. Line 125: please erase enzyme

Reply: Erased

13. Line 150: please change blue lines with black frames

Reply: Changed as “…. are indicated by red and black frames, respectively.”

14. Line 205: please specify that site-1 is in the allosteric pocket (the same for site-2)

Reply: We have specified this in revised version.

15. The Section 3 should dedicate to the discussion of results as done, however the authors introduce again the topic of the research in an extensive way. Please revise this part. According to me the paragraph from line 292 to 313 can be removed and if necessary integrate it in the introduction, without repeating the concepts.

Reply: As suggested, the paragraph from line 292 to 313 has been removed in the revised manuscript.

16. I think the differences between the data presented in the manuscript for α-glucosidase (lines 326-329) and those previously reported can be related with the enzyme and substrate concentrations, but not with the base concentration. Another variable could be the different purity of tested compounds.

Reply: It has been revised accordingly.

17. Line 340: please change bis form with “a dimeric form”

Reply: “bis-form” has been replaced with “dimeric form.”

18. Line 348: please modify the sentence as: “In order to have information on the mechanism …”

Reply: The sentence has been revised.

19. Line 352: the sentence “In non-competitive enzyme inhibition Kic = Kiu” makes no sense in this form, it can be omitted.

Reply: As suggested, that sentence no longer exists in the revised version.

In addition to this, we have revised our manuscript according to some comments from other reviewers. We would really appreciate if you have any comments in the revised version.

Reviewer 2 Report

Manuscript of interest with good results well depicted.

no modifications needed

Author Response

We would like to express our sincere thanks to the Reviewer 2 for the positive comment (in italics) on our paper.

Reviewer 2

Manuscript of interest with good results well depicted.

no modifications needed

Reply: We thank the reviewer for positive response. However, we have modified our manuscript in response to other reviewers. So, we would really appreciate your comments and suggestions in the revised version if have any.

Reviewer 3 Report

In this manuscript authors have isolated and studied 2,3,6-Tribromo-4,5-Dihydroxybenzyl Derivatives from Symphyocladia latiuscula for anti-diabetic activity. They have studied the mechanism of PTP1B and α-glucosidase enzymes inhibition by bromophenols (1-3) by kinetic analysis and molecular docking simulation.

The study is interesting and useful but I do not see any originality in the work. I would highly recommend to add in vivo study data to support the in vitro studies.

In discussion section, author have compared PTP1B inhibitory activity of their compound with the studies done by Liu et. al published in 2011. The 2-fold improvement or changes in inhibitory concentration is not considered significant.

Why have authors not used Rosiglitazone as a reference drug for the PTP1B inhibitory activity comparison in table 2?

In the experimental data section, authors have not reported the molecular weight and MS data for compound 1 and 2.

There is mistake in the NMR data of the compound 3. Page 13, line 455; replace 2H by 4H.

Author Response

We express our sincere thanks to the Reviewer 3 for constructive comments and suggestions. We have revised our manuscript in the light of those comments and point-by-point responses to the Reviewer comments are provided below (with Reviewer comments in italics).

Reviewer 3

In this manuscript authors have isolated and studied 2,3,6-Tribromo-4,5-Dihydroxybenzyl Derivatives from Symphyocladia latiuscula for anti-diabetic activity. They have studied the mechanism of PTP1B and α-glucosidase enzymes inhibition by bromophenols (1-3) by kinetic analysis and molecular docking simulation.

The study is interesting and useful but I do not see any originality in the work. I would highly recommend to add in vivo study data to support the in vitro studies.

Reply: We do agree with the in vivo study recommendation. At this moment, amount of test compounds is not sufficient enough to conduct in vivo study. We have initiated the isolation process to obtain a good amount of compounds in hand. In support of findings of the present in vitro and in silico study, we have planned in vivo study along with some GPCRs functional assays that will be reported in a forthcoming article.

In discussion section, author have compared PTP1B inhibitory activity of their compound with the studies done by Liu et. al published in 2011. The 2-fold improvement or changes in inhibitory concentration is not considered significant.

Reply: It is not reliable to compare the results from different studies and state the significance in activity. PTP1B inhibitory activity of one of three compounds of present study had ever been reported which was nearly 2-fold different in activity, and we have discussed the possible reasons for this variation in discussion section. However, simultaneous evaluation of PTP1B inhibitory activity of three bromophenols in the present study showed significant differences in their IC50 values.

Why have authors not used Rosiglitazone as a reference drug for the PTP1B inhibitory activity comparison in table 2?

Reply: Ursolic acid is a natural pentacyclic triterpenoid that is commonly used as a reference compound for PTP1B enzyme inhibition assay. So we used this reference compound to validate and compare the result of PTP1B inhibition in the present study. However, rosiglitazone is a FDA approved thiazolidinedione drug that improves insulin sensitivity in T2DM. In that regard, we analyzed the insulin sensitizing potential of bromophenols 1-3 and rosiglitazone in IR HepG2 cells to validate and compare the result, and put forth the action mechanism.

In the experimental data section, authors have not reported the molecular weight and MS data for compound 1 and 2.

Reply: We have included the molecular weight and MS data for compound 1 and 2 in the revised version.

There is mistake in the NMR data of the compound 3. Page 13, line 455; replace 2H by 4H.

Reply: We apologize for this error. We have corrected this in the revised version.

Reviewer 4 Report

The novelty of this manuscript is that the authors of the present article explore for the first time the mechanism of enzyme inhibition by three bromophenols on tyrosine phosphatase 1B (PTP1B) and α-glucosidase via enzyme kinetics and molecular docking studies. In particular, as they reported in the manuscript, in order to predict the binding affinity of these ligands to an enzyme’s active site, and to understand the binding mechanism, molecular docking simulations are performed. The obtained results could be very interesting and could be useful for the development of new anti-diabetic drugs starting from these natural compounds. However, some parts of the molecular modelling studies should be clarified and improved.

1) The authors wrote at the beginning of page 7 that “To elucidate the interaction between bromophenols and α-glucosidase (PDB ID: 3A4A), docking simulations were performed using AutoDock 4.2.” 3A4A is the crystal structure of isomaltase from Saccharomyces cerevisiae. Then, the authors have to give an explanation on this point if they used directly this x-ray structure or, as reported in reference 56, due to high sequence homology, they built a homology model of α-glucosidase starting from this structure and then, they used this model for the docking studies. It is very important that the enzyme used for the docking simulations is the same used for experimental studies in order to correlate the theoretical and the experimental results.

2)The authors wrote at page 12 “Docking simulations for α-glucosidase revealed the involvement of two sites (site-1 and site-2) in the allosteric pocket. Ding et al. [45] predicted five binding cavities on α-glucosidase, one of which is an active pocket and the other four of which are allosteric pockets away from the active site that bind non-competitive inhibitors. Our results suggest that 1 and 2 are non-competitive inhibitors that bind to site-1 and site-2 in the allosteric pocket. Our results also indicate that 3 is a mixed mode inhibitor that binds one bromophenol ring to the active pocket and the other bromophenol ring to site-2 in the allosteric pocket.” It is not clear the criteria of selection used by authors for these binding modes. The authors reported only the binding energy of the selected complexes in the table 4 but they have to explain how they selected, as most probable binding mode, the site-1 and site-2 with respect to the other two allosteric binding cavities.

3)In the paragraph “Molecular docking simulation of PTP1B inhibition”, at page 5, the authors have to describe also the main interactions of bromophenols 1 and 2 within the catalytic site and the results obtained with the reference compounds. This part is missing and it is important for the discussion of their results. They presented these results only in the table 3 and figure 5.

4)For the same reason, in the paragraph “Molecular Docking Simulation of α-Glucosidase Inhibition” the authors have to describe the results obtained with the reference compound C.

5)The discussion at the end of page 11 is not clear. The authors in this point of the manuscript correlated their results on PTP1B with those present in the literature but they cited not only the residues identified in the binding sites of the considered bromophenols and they did not discriminate between the residues present in the active and allosteric site. Since this part is very important to understand the binding mechanism of the bromophenols and to evidence the residues which play a key role in their mechanism of inhibition, the authors have to rewrite this part considering also the results obtained with the reference compounds.  Accordingly, the authors have to discuss in the same way the results obtained on α-glucosidase. This discussion is missing.

In conclusion, in my view, this paper is to be published on Marine Drugs only after these modifications.

Author Response

We appreciate for the constructive comments and suggestions by Reviewer 4. We have revised the manuscript according to the suggestions and point-by-point responses to the Reviewer comments are provided below (with Reviewer comments in italics).

Reviewer 4

The novelty of this manuscript is that the authors of the present article explore for the first time the mechanism of enzyme inhibition by three bromophenols on tyrosine phosphatase 1B (PTP1B) and α-glucosidase via enzyme kinetics and molecular docking studies. In particular, as they reported in the manuscript, in order to predict the binding affinity of these ligands to an enzyme’s active site, and to understand the binding mechanism, molecular docking simulations are performed. The obtained results could be very interesting and could be useful for the development of new anti-diabetic drugs starting from these natural compounds. However, some parts of the molecular modelling studies should be clarified and improved.

1) The authors wrote at the beginning of page 7 that “To elucidate the interaction between bromophenols and α-glucosidase (PDB ID: 3A4A), docking simulations were performed using AutoDock 4.2.” 3A4A is the crystal structure of isomaltase from Saccharomyces cerevisiae. Then, the authors have to give an explanation on this point if they used directly this x-ray structure or, as reported in reference 56, due to high sequence homology, they built a homology model of α-glucosidase starting from this structure and then, they used this model for the docking studies. It is very important that the enzyme used for the docking simulations is the same used for experimental studies in order to correlate the theoretical and the experimental results.

Reply: Since, the 3D structure of α-glucosidase used in biological assays from yeast has not been reported yet, isomaltase from Saccharomyces cerevisiae co-crystallized with maltose (PDB ID: 3A4A) was used as the α-glucosidase protein because it shows 85% similarity to yeast α-glucosidase (MAL12) through homology modeling [28]. We have included this explanation in the respective section (Page 7).

2)The authors wrote at page 12 “Docking simulations for α-glucosidase revealed the involvement of two sites (site-1 and site-2) in the allosteric pocket. Ding et al. [45] predicted five binding cavities on α-glucosidase, one of which is an active pocket and the other four of which are allosteric pockets away from the active site that bind non-competitive inhibitors. Our results suggest that 1 and 2 are non-competitive inhibitors that bind to site-1 and site-2 in the allosteric pocket. Our results also indicate that 3 is a mixed mode inhibitor that binds one bromophenol ring to the active pocket and the other bromophenol ring to site-2 in the allosteric pocket.” It is not clear the criteria of selection used by authors for these binding modes. The authors reported only the binding energy of the selected complexes in the table 4 but they have to explain how they selected, as most probable binding mode, the site-1 and site-2 with respect to the other two allosteric binding cavities.

Reply: The docking simulation was conducted with 15 independent genetic algorithms (GAs) with the default parameters. In case of 1, all fifteen complexes were docked into the site-1 of allosteric pocket like (Z)-3-butylidenephthalide (compound D), a reference allosteric inhibitor. In case of 2, all fifteen complexes were docked into the site-2 of allosteric pocket. However, in case of 3, seven of fifteen enzyme-3 complexes were docked into the site-2 of allosteric pocket, while remaining eight complexes were docked into both the catalytic pocket and site-2 of allosteric pocket with lowest binding energy. Therefore, we concluded that 3 is mixed-type inhibitor which bind both catalytic and site-2 of allosteric pocket.

3)In the paragraph “Molecular docking simulation of PTP1B inhibition”, at page 5, the authors have to describe also the main interactions of bromophenols 1 and 2 within the catalytic site and the results obtained with the reference compounds. This part is missing and it is important for the discussion of their results. They presented these results only in the table 3 and figure 5.

Reply: It has been revised and the missing part has been included as “The PTP1B-1 inhibitor complex at catalytic pocket showed four H-bond interactions (Lys116, Asp181, Arg221 and Gln226) at a distance of 2.0 to 2.2 Å. In addition, bromine atoms at C2, C3 and C6 position involved in multiple interactions with Thr263, Val184, Gln266, Gly183, Arg221, Phe182, Lys116 and Trp176. Similarly, PTP1B-2 inhibitor complex displayed three H-bond interactions with Arg221, Gln266 and Gly183 at a distance of 1.9 to 2.1 Å. Three bromine atoms of 2 showed multiple interactions with Arg211, Gln266, Trp179, Thr283, Asp265, Asp181, Lys116, Phe182 and Val184. The catalytic inhibitor, compound A, formed a complex with PTP1B at catalytic site forming four H-bond interactions with Asp48, Lys116, Lys120 and Asp181 along with other non-polar interactions with Ala217, Tyr46, Met258 and Gln262. Common interacting residues for inhibitors (1, 2 and compound A) at catalytic pocket involved Asp181 and Lys116, which are determining residues for catalytic inhibition.”

4)For the same reason, in the paragraph “Molecular Docking Simulation of α-Glucosidase Inhibition” the authors have to describe the results obtained with the reference compound C.

Reply: Results obtained with reference compounds has been described in the revised version as “The optimal conformation of 1 was inside site-1 in the allosteric pocket with the lowest energy of −5.94 kcal/mol, forming H-bond interactions with Thr290 and Glu271 at a distance of 2.2 Å. Non-polar interactions were formed with Asp341, Cys342, Ser296, Glu271, Lys16 and Lys13 (Figure 6A). Bromophenol 2 was most appropriately suited to bind in site-2 in the allosteric pocket with the lowest binding energy of −5.61 kcal/mol, forming H-bond interactions with Tyr158, Arg315, Glu411 and Asn415 located at a distance 2.2 Å. Non-polar interactions were formed with Arg315, Lys56, Gly160, Phe314 and Leu313 (Figure 6B). The reference allosteric inhibitor (Z)-3-butylidenephthalide (Compound D) bound to site-1 of the allosteric pocket with a binding energy of −6.85 kcal/mol, forming two H-bond interactions with Glu296 and His295 along with several non-polar interactions with Trp15, Lys16, Asn259, Ala292, Thr290, Arg294, Leu297, Ser291, Ser298, Asp341, Cys342 and Trp343 at the allosteric site. However, compound C (reference catalytic inhibitor) demonstrated numerous H-bond interactions (Asp69, His112, Tyr158, Gln182, Asp215, Arg213, Ser240, Asp242, Glu277, His280, Asp307, Asp352 and Arg442) at the catalytic pocket along with non-polar interactions involving Tyr72, Lys156, Phe178, Val216, Gln279, Phe303, Arg315, Gln353, His351, Glu411. Interestingly, 3 showed a mixed mode of inhibition by interacting with both catalytic and allosteric regions (specifically at site 2). Since 3 is a dimer that comprises two bromophenol groups, one of the bromophenol groups bound to the active catalytic pocket and the other bound to the allosteric pocket at site-2 (Figure 4B). Hydroxyl groups at the C-4 and C-5 positions of ring A formed two H-bonds with Ser157 at a distance of 2.1 and 2.3 Å. Non-polar interactions between the bromine atom and Asn415, Ser157, Tyr156 and Arg315 were also observed at the catalytic pocket (Figure 6C). However, at site-2 in the allosteric pocket, the -OH groups at C-4´and C-5´ displayed H-bond interactions with Asp307 and Gln353 located at a distance of 1.9 and 2.2 Å, respectively. Non-polar interactions were observed between bromine atoms and His280, Phe303, Gln279, Arg315, Tyr158 and Asp307 at site-2 in the allosteric pocket.”

5)The discussion at the end of page 11 is not clear. The authors in this point of the manuscript correlated their results on PTP1B with those present in the literature but they cited not only the residues identified in the binding sites of the considered bromophenols and they did not discriminate between the residues present in the active and allosteric site. Since this part is very important to understand the binding mechanism of the bromophenols and to evidence the residues which play a key role in their mechanism of inhibition, the authors have to rewrite this part considering also the results obtained with the reference compounds.  Accordingly, the authors have to discuss in the same way the results obtained on α-glucosidase. This discussion is missing.

In conclusion, in my view, this paper is to be published on Marine Drugs only after these modifications.

Reply: As suggested, discussion part has been revised as follows.

“Result of PTP1B docking revealed that 1 and 2, and reference allosteric inhibitor (compound B) shared common interacting residues Ala189, Asn193, Glu266, Leu192, Phe196 and Phe280 at the allosteric site. These are prime interacting residues at the allosteric site, and 3 did not show interactions with any of these residues. Instead, it showed multiple bonding with prime amino acid residues Asp181, Arg221 and Lys116, that were in common with the reference catalytic inhibitor (compound A). Interestingly, 1 and 2 showed additional interactions with Arg221 and Lys116 at the catalytic site, representing mixed-type inhibition. The H-bond interactions of 1-3 and reference inhibitors at respective binding sites are similar in number, however, bromine atoms of 1-3 showed numerous non-polar interactions which aids in enhancing stability at the binding sites.

Docking simulations for α-glucosidase revealed the involvement of two sites (site-1 and site-2) in the allosteric pocket. The docking simulation was conducted with 15 independent genetic algorithms (GAs) with the default parameters. In case of 1, all fifteen complexes were docked into the site-1 of allosteric pocket like (Z)-3-butylidenephthalide (compound D), a reference allosteric inhibitor. In case of 2, all fifteen complexes were docked into the site-2 of allosteric pocket. However, in case of 3, seven of fifteen enzyme-3 complexes were docked into the site-2 of allosteric pocket, while remaining eight complexes were docked into both the catalytic pocket and site-2 of allosteric pocket with lowest binding energy. Therefore, we concluded that 3 is mixed-type inhibitor which bind both catalytic and site-2 of allosteric pocket. Ding et al. [43] predicted five binding cavities on α-glucosidase, one of which is an active pocket and the other four of which are allosteric pockets away from the active site that bind non-competitive inhibitors. Similarly, Didem et al. [44] had reported details of the sitemap results for α-glucosidase enzyme from Saccharomyces cerevisiae, including binding residues at catalytic site and four allosteric binding sites (ABS1-ABS4). In our docking result, allosteric inhibition involved only two different sites, which we named as site-1 and site-2. Bromophenol 1 showed allosteric inhibition by binding to site-1 of allosteric pocket via various H-bond and non-polar interacting residues (Table 4) of which Glu271 and Ala292 are common residues at ABS4. Similarly, 2 bound to different site other than site-1, and we named it as site-2. Interacting residues at site-2 that involved in binding of 2 were Tyr158, Asn415, Lys156, Gly160 and Phe314. These residues were reported to interact at ABS1 allosteric site. The reference allosteric inhibitor (Z)-3-butylidenephthalide (compound D) and 1 demonstrated the interaction with common residues Thr290, Lys16, Ala292, Cys342, Ser298, Leu297 and Trp343, demonstrating that these two bound at the same site (site-1). These results suggest that 1 and 2 are non-competitive inhibitors that bind to site-1 and site-2 in the allosteric pocket. Acarbose (compound C) is a reference catalytic inhibitor that bound to catalytic pocket with lowest binding energy and demonstrated the involvement of H-bond interacting residues Asp69, His112, Gln182, Asp215, Arg213, Ser240, Asp242, Glu277, His280, Asp307, Asp352 and Arg442, and non-polar interactions with Tyr72, Lys156, Phe178, Val216, Gln279, Phe303, Gln353, His351 and Glu411. Interestingly, 3 showed interactions with Asp307, His280, Asp352, Gln353, Gln279 and Phe303 at the catalytic site similar to acarbose, and with Tyr158, Asn415 and Arg315 common to 2 at site-2 of allosteric pocket. These results also indicate that 3 is a mixed mode inhibitor that binds one bromophenol ring to the active pocket and the other bromophenol ring to site-2 in the allosteric pocket (Figure 6C). In addition, non-polar interactions between bromine and Tyr158 and Arg315 further stabilize the binding of 3 to both binding pockets. The overall results of our docking simulation and enzyme kinetics analysis confirm that the allosteric inhibition observed for this enzyme is due to non-competitive binding. This is in line with research that reports that allosteric inhibitors generally function through non-competitive or mixed binding [45]. It is notable that compounds with more bromine atoms and/or bromophenol rings have increased activity, which is likely due to the involvement of a large number of non-polar interactions that play a vital role in the positioning of inhibitors into the active site with a high degree of stability.”

Round 2

Reviewer 3 Report

Dear Authors,

I am glad to see the significant improvement in the manuscript. I understand that the amount of product for in vivo studies is always a concern especially when it is isolated from natural resources but I still emphasize on adding the data when available.

Reviewer 4 Report

The authors improved or corrected the manuscript. However, this manuscript is to be published on Marine Drugs only after these modifications. They have to modify the sentence "The structure of α-glucosidase with its catalytic ligand α-D-glucose (PDB ID: 3A4A) and the structure of acarbose (known as ‘Compound C’ in our study) and (Z)-3-butylidenephthalide (BIP) (known as ‘Compound D’ in our study) were obtained from the RCSB Protein Data Bank…." in the paragragh "4.8 Molecular Docking Simulation of PTP1B and α-Glucosidase Inhibition". In particular, they have to introduce the procedure performed to build the homology model of α-glucosidase starting from the  structure of isomaltase from Saccharomyces cerevisiae (PDB ID: 3A4A). Moreover, they have to correct in different parts of the manuscript the sentence "α-glucosidase (PDB ID: 3A4A)" since the PDB 3A4A is the structure of isomaltase from Saccharomyces cerevisiae.